# Differential Immunostimulatory Effects of Hydrophilic and Hydrophobic *Solanum trilobatum* Fractions in Tilapia

**DOI:** 10.3390/biology14101333

**Published:** 2025-09-27

**Authors:** M. Divya Gnaneswari, D. Christybapita, Smriti Sharma, Shivani Tyagi, R. Dinakaran Michael, Parasuraman Aiya Subramani

**Affiliations:** 1Centre for Fish Immunology, P.G. Department of Zoology and Biotechnology, Lady Doak College, Madurai 625002, India; m.divya.gnaneswari@gargi.du.ac.in (M.D.G.);; 2Department of Zoology, Gargi College, Delhi University, New Delhi 110049, India; smriti.sharma@gargi.du.ac.in (S.S.); shivani.tyagi@gargi.du.ac.in (S.T.); 3Thünen Institute of Fisheries Ecology, Herwigstraße 31, 27572 Bremerhaven, Germany

**Keywords:** *Oreochromis mossambicus*, *Solanum trilobatum*, immunostimulant, *Aeromonas hydrophila*, non-specific immunity, antibody response

## Abstract

Disease outbreaks in fish farms, especially under crowding and confinement, are responsible for causing major economic loss to farmers. This study has been designed to examine the ability of two extracts (water or hexane soluble fraction, WSF or HSF) from the leaves of the plant *Solanum trilobatum* to enhance the health of tilapia fish. The fish were fed with these extracts incorporated into their diet for 1, 2, or 3 weeks. Results showed that both extracts improved the immune system over time by enhancing globulin level, lysozyme, and anti-protease activity. HSF enhanced key immune responses like reactive oxygen species (ROS) and myeloperoxidase activity (MPO) and led to higher antibody levels than WSF. When the fish were challenged with *Aeromonas hydrophila*, those fed HSF had lower death rates at all time points, while WSF only helped at certain doses and days. Chemical analysis showed HSF had high levels of aromatic compounds and phytosterols which are likely easier for the fish to absorb. WSF had smaller alcohol and carbonyl compounds, but in lower amounts. Because of better absorption, HSF was more effective at strengthening immunity and protecting fish from disease.

## 1. Introduction

The Blue transformation roadmap [1] aims to strengthen aquatic food system by promoting nutritious food security, sustainable employment, environmental restoration, and socio-economic development. Central to this vision is the emphasis on efficient aquaculture production and improved living conditions for farmed fish, addressing growing consumer concerns. A key objective highlights the need for effective stock management, which is increasingly challenged by intensive fish farming practices that expose fish to multiple stressors, compromising their health and increasing susceptibility to infectious agents.

One of the critical strategies outlined in the roadmap is the prophylactic control of disease through immune system stimulation using immunostimulants, rather than relying on potentially hazardous chemotherapeutics and antibiotics [2]. Overuse of such chemical treatment’s risks promoting antibiotic-resistant pathogens, bioaccumulation in tissues, and environmental contamination. Although vaccines represent a widely used and effective prophylactic measure, they remain costly for many fish farmers and do not provide broad-spectrum protection against diverse pathogens [3].

Immunostimulants from various categories have been tested in fish, demonstrating their ability to enhance immunity, protect against diseases, and have minimal environmental impact, thus ensuring the safety of fishery products for consumers. Potential immunostimulants tested in fish include chemicals, bacterial extracts, algal products, as well as animal and plant extracts. A comprehensive review of immunostimulant research across different continents can be found in Subramani et al. [4].

*Solanum trilobatum* (Family: Solanaceae) commonly known as purple-fruited pea eggplant or “Thoothuvalai” in Tamil, is rich in bioactive phytochemicals (alkaloids, flavonoids, tannins, glycosides, simple phenols, phenolic acids, isoflavones, xanthones, and lignans). It contains alkaloids such as solasodine, which exhibit antimicrobial [5] and anti-inflammatory properties [6]. Flavonoids, tannins, and glycosides are also present, contributing to its antioxidant potential [7]. The leaves and berries are notable for their high content of phenolic compounds and essential amino acids. Additionally, it contains trace elements like iron, calcium, and phosphorus that support its traditional medicinal use in respiratory ailments [8]. In our previous study, intraperitoneal injection of water-soluble and hexane-soluble fractions of *Solanum trilobatum* leaves in *Oreochromis mossambicus* enhanced nonspecific immune mechanisms, including serum lysozyme activity and the production of reactive oxygen and nitrogen species. These fractions also reduced mortality rates following a challenge with live virulent *Aeromonas hydrophila* [9].

Previous studies have identified various bioactive compounds in *S. trilobatum*, with key constituents including Epoxylinalol, Himachalol, Illudol, Epibuphanamine, Baimuxinal, and Edulan IV, as determined by gas chromatography–mass spectrometry (GC-MS). Additionally, nonpolar solvent extraction has revealed the presence of simple phenols, phenolic acids, isoflavones, xanthones, and lignans, as detected using thin-layer chromatography [7]. One of the primary limitations of natural product efficacy in vivo is bioavailability. While polar compounds dissolve readily in body fluids and are rapidly excreted, nonpolar compounds tend to persist longer in the system, increasing their potential bioactivity [10]. Drug candidates have traditionally been selected based on their hydrophobicity, which influences both absorption and retention [11].

This study evaluates the immunostimulatory efficacy of water-soluble (WSF) and hexane-soluble (HSF) fractions of *S. trilobatum* administered through feed for 1, 2, and 3 weeks in tilapia (*O. mossambicus*). Immunological parameters assessed include non-specific immune responses (serum globulin levels, lysozyme activity, antiprotease activity, reactive oxygen species [ROS], reactive nitrogen intermediates [RNI], and myeloperoxidase [MPO] production in peripheral blood leukocytes) as well as the specific immune response (antibody production against *A. hydrophila*). Additionally, the ability of these feed supplements to enhance disease resistance was evaluated through an experimental *A. hydrophila* challenge.

GC-MS analysis revealed that HSF contained a higher proportion of aromatic compounds (benzenoids) and steroids, mainly phytosterols, while WSF primarily consisted of short-chain alcohol and carbonyl compounds in lower amounts. Due to their hydrophobic nature and higher bioavailability, aromatic and steroid compounds likely contributed to the superior immunostimulatory effects of HSF, enhancing both nonspecific and specific immunity and improving disease resistance. These findings support the use of *S. trilobatum* HSF as an effective feed supplement for tilapia aquaculture.

## 2. Materials and Methods

### 2.1. Fish and Maintenance

Male *O. mossambicus* were obtained from a local farmer and were housed in fibre-reinforced plastic (FRP) tanks at a stocking density of 4 g/L kept under ambient temperature 28 ± 2 °C and natural light conditions. Fish weighing 25 ± 5 g and measuring 11.63 ± 0.12 cm in length were used for serological assays, while fish weighing 50 ± 5 g and measuring 14.29 ± 0.21 cm were used for cellular assays. Water in the tanks was partially replaced every other day to maintain hygiene and stability. Key water quality parameters were monitored regularly: pH was 7.3 ± 0.3 and dissolved oxygen was 5.2 ± 0.1 mg/L. A handheld pH meter was used to measure water pH and dissolved oxygen was estimated using Winkler’s titration method. Before the experiment began, fish were acclimated for two weeks and fed a balanced, lab-prepared diet *ad libitum*.

### 2.2. Plant Extract Preparation and Experimental Design

Leaves of *S. trilobatum* L. (Family: Solanaceae) were processed to obtain water-soluble (WSF) and hexane-soluble fractions (HSF) following earlier protocols [9]. These extracts were mixed into the fish feed (supplemented feed) at concentrations of 0.01%, 0.1%, and 1% of total feed weight. The dose selection was based on the results of intraperitoneal administration of the same immunostimulant. Control fish received an unsupplemented balanced diet (protein: 39%, carbohydrate: 24%, lipid: 11%, ash: 9%) and were fed daily at 2% of their body weight for three weeks. At the end of each week, six fish per group were randomly selected and bled via the common cardinal vein [12], adhering to the guidelines [https://ccac.ca/Documents/Standards/Guidelines/Fish.pdf, accessed on 25 September 2025]. Serum from collected blood samples were separated and stored at –20 °C in sterile microfuge tubes. Leukocytes were isolated from peripheral blood using Lymphosep (ICN biomedicals, Irvine, CA, USA) density gradient for ROS, RNI, and MPO assays, following previously published methods [9].

### 2.3. Nonspecific Immune Response

Serum total protein was estimated by following the method of Lowry et al. [13], and albumin levels were determined through the bromocresol green (BCG) dye-binding method [14]. By subtracting albumin from total protein, the globulin content was determined.

Serum lysozyme activity was determined using the method described by Hutchinson et al. [15]. *Micrococcus lysodeikticus* (Sigma Aldrich, Bengaluru, India) was used as the substrate, and one unit of activity was defined as a decline in absorbance of 0.001 per minute.

Antiprotease activity in serum was assessed following the method of Bowden et al. [16] using Na-benzoyl-DL-arginine-p-nitroanilide (BAPNA, HiMedia, Mumbai, India) as a substrate. Trypsin-mediated hydrolysis produced p-nitroaniline, measured colorimetrically. The percentage of trypsin inhibition was calculated as per Zuo and Woo [17] using the formula$$Percentage\;trypsin\;inhibition=\left(1-\frac{\;Sample\;OD}{Trypsin\;blank\;OD}\right)\times 100$$

The production of superoxide anions by leukocytes was estimated using the nitroblue tetrazolium (NBT, HiMedia, Mumbai, India) reduction assay. The formation of insoluble formazan indicated the level of superoxide activity, and the solubilized product was quantified using a microplate reader (Bio-Rad, Hercules, CA, USA) [18].

Nitric oxide (NO) production by peripheral blood leukocytes was quantified using the Griess reaction. Stable nitrite in the supernatant was measured colorimetrically after converting to pink azo dye using Griess reagent [19]. Nitrite concentration (NO_2_^−^) was calculated from a standard curve generated using known concentrations of sodium nitrite.

Myeloperoxidase (MPO) content was determined according to Palić et al. [20], with slight adjustments. Neutrophils in head kidney leukocytes were lysed using cetyltrimethylammonium bromide (CTAB, HiMedia, Mumbai, India) to release MPO, which catalysed the oxidation of TMB (3,3′,5,5′-tetramethylbenzidine, Genei, Bengaluru, India) in the presence of hydrogen peroxide and absorbance was taken at 450 nm to quantify the reaction.

### 2.4. Specific Immune Response

A virulent strain of *Aeromonas hydrophila* (AHO21) was obtained from Prof. M. R. Chandran, Department of Animal Science, Bharathidasan University, Tiruchirappalli, India. The culture was grown in tryptic soy broth (HiMedia, Mumbai, India), and heat-killed bacterial cells were prepared by adjusting the concentration with phosphate-buffered saline (PBS) as per Karunasagar et al. [21].

Fish were divided into two sets, each containing ten groups. One group in each set served as control and received a regular diet. The remaining nine groups were fed with diets supplemented with either WSF or HSF of *S. trilobatum* leaves at 0.01%, 0.1%, or 1% for one, two, or three weeks. At the end of each feeding duration, fish were immunized with the heat-killed *A. hydrophila*. Blood samples were collected at seven-day intervals using sterile serological tubes (70 × 10 mm) and the serum separated was stored at −20 °C in sterile microfuge tubes until used for antibody analysis.

Antibody levels specific to *A. hydrophila* were measured using an indirect ELISA method based on Binuramesh et al. [22]. After immunization, fish were maintained on a regular control diet during the antibody monitoring period.

### 2.5. Disease Resistance

*A. hydrophila* cells were harvested by centrifugation at 800× *g* for 15 min, washed with PBS, and resuspended to the required dose for challenge studies. Fish (*n* = 10 per group, in triplicate) were fed diets supplemented with WSF or HSF and control groups received an unsupplemented balanced diet. Feeding schedules were grouped into Set I (1 week), Set II (2 weeks), and Set III (3 weeks). After the respective feeding periods, fish were challenged with live *A. hydrophila* (1 × 10^8^ cells/fish).

Fish were observed for 15 days post-challenge, and daily mortalities were recorded. Clinical signs included haemorrhagic septicemia, swollen abdomen, and external lesions on the ventral surface. The cause of mortality was confirmed by re-isolating *A. hydrophila* from liver samples of 10% of the dead fish, using *Aeromonas* isolation medium (HiMedia, Mumbai, India). Following the challenge, the control diet was given to every fish. In accordance with Ellis [23], relative percent survival (RPS) was computed using the following formula:$$RPS=\left(1-\;\frac{Percent\;mortality\;in\;treated\;group}{Percent\;mortality\;in\;control\;group}\right)\times 100$$

### 2.6. GC-MS Analysis

The phytochemical analysis of both the fractions of *S. trilobatum* leaves was carried out using a GC-MS-QP 2010 system (Shimadzu, Kyoto, Japan) equipped with a thermal desorption system (TD-20). The analytical procedure followed the protocol of Sharma et al. [24], with minor modifications. Operating conditions included injection temperature of 260 °C, ion source temperature of 220 °C, and interface temperature of 270 °C. The initial column oven temperature and flow parameters were maintained as reported in the previous report [25].

Each compound detected was identified by comparing its retention time and mass spectrum with reference spectra available in the Wiley and NIST (National Institute of Standards and Technology) libraries. The relative abundance of each constituent was expressed as a percentage of the total peak area.

### 2.7. In Silico Evaluation of Bioavailability of the Phytoconstituents

To evaluate the oral bioavailability of the major phytoconstituents in the WSF and HSF extracts of *S. trilobatum*, a computational analysis was carried out using the Molinspiration Cheminformatics tool (https://www.molinspiration.com). The molecular structures of the compounds were obtained from the PubChem database [26] in SMILES (Simplified Molecular Input Line Entry System) format. These SMILES entries were uploaded into the Molinspiration platform to generate physicochemical descriptors relevant to drug-likeness and bioavailability.

Key parameters assessed included molecular weight (MW), predicted lipophilicity (miLogP), topological polar surface area (TPSA), number of hydrogen bond donors (nOHNH), and number of hydrogen bond acceptors (nON). The values were interpreted in relation to Lipinski’s Rule of Five, which suggests that compounds are likely to show good oral bioavailability if they meet the following criteria: MW ≤ 500 Da, miLogP ≤ 5, nOHNH ≤ 5, nON ≤ 10, and TPSA < 140 Å^2^.

### 2.8. Statistical Analysis

SigmaStat (4.0) was used to plot and to conduct the statistical analyses. The data were expressed as the arithmetic mean ± standard error (SE). Statistical analysis of the data involved a one-way analysis of variance (ANOVA) followed by Tukey’s pairwise comparison test. Prior to conducting ANOVA, the normal distribution of the data was assessed using the Shapiro–Wilk test, and homogeneity of variances was evaluated using Bartlett’s test. The levels of significance were expressed as *p*-values less than 0.05, indicating significant difference within a week, as provided in the graph with different letters.

## 3. Results

### 3.1. Nonspecific Immune Response

#### 3.1.1. Globulin Level

Feeding with 0.1% or 1% of the water-soluble fraction (WSF) significantly increased serum globulin levels after one week (*p* < 0.05; Figure 1a). The group receiving 1% of the hexane-soluble fraction (HSF) for one week showed the highest globulin levels (Figure 1b). All WSF doses elevated globulin only after three weeks, while all HSF doses caused an increase after two weeks, which persisted into the third week.

#### 3.1.2. Lysozyme Activity

A significant enhancement in lysozyme activity was observed in fish fed with 1% WSF for one or two weeks (*p* < 0.05) which is significantly higher than HSF. Feeding with 0.1% WSF also raised lysozyme levels after one week (Figure 2a). No change was seen in the group fed with 0.01% WSF.

In HSF-fed groups (Figure 2b), 0.01% HSF led to a significant rise in lysozyme activity after three weeks, and 0.1% HSF induced an increase after two and three weeks. However, 1% HSF did not show any notable effect across all durations, and none of the HSF doses altered lysozyme activity after just one week.

#### 3.1.3. Antiprotease Activity

Supplementation with WSF for 1–3 weeks improved serum antiprotease activity in the 0.1% and 1% groups, notably after one week (*p* < 0.05; Figure 3a). The 0.01% WSF did not significantly affect activity till 2 weeks of feeding.

In contrast, all doses of HSF elevated antiprotease levels after one week of feeding (*p* < 0.05; Figure 3b). However, this effect did not persist with extended feeding durations except in the group fed with 1% HSF for 3 weeks. In comparison, 0.01% of WSF and 1% HSF-incorporated diet enhanced the activity more effectively after three weeks of feeding than any other dose and any other week.

#### 3.1.4. Reactive Oxygen Species (ROS) Production in Peripheral Blood Leucocytes

As shown in Figure 4a, 0.01% WSF significantly increased reactive oxygen species (ROS) production after one, two, and three weeks of feeding (*p* < 0.05), with levels rising progressively over time. A similar increase was observed after two and three weeks in the 0.1% WSF group, while 1% WSF induced a significant effect only after three weeks.

For HSF-fed fish (Figure 4b), 0.01% supplementation significantly enhanced ROS levels at all three time points (*p* < 0.05). The 0.1% and 1% HSF doses showed increased activity only after two or three weeks of feeding.

#### 3.1.5. Reactive Nitrogen Intermediate (RNI) Production in Peripheral Blood Leucocytes

Dietary WSF at 0.01%, significantly boosted reactive nitrogen intermediate (RNI) production after one week (*p* < 0.05; Figure 5a). The 1% WSF group showed elevated RNI levels after two and three weeks, with the peak observed at week three.

In HSF-treated groups (Figure 5b), all doses significantly increased RNI production after three weeks (*p* < 0.05). Additionally, 0.01% and 1% HSF also enhanced RNI levels after just one week.

#### 3.1.6. Myeloperoxidase Activity in Peripheral Blood Leucocytes

The 0.01% and 1% WSF groups showed increased myeloperoxidase (MPO) activity after two weeks of feeding (Figure 6a), with the highest level recorded in the 1% group after one week. However, WSF did not significantly modulate MPO levels after three weeks.

HSF supplementation, on the other hand, resulted in a sustained increase in MPO activity after both two and three weeks of feeding (Figure 6b).

### 3.2. Specific Immune Response

In WSF-fed fish, antibody titres increased significantly on day 7 across all treatment groups after one week of feeding (*p* < 0.05; Figure 7a). In the 1% WSF group, this response remained elevated into the third and fourth weeks. HSF-treated groups also showed a consistent antibody increase after one week (Figure 7b).

After two weeks, the 1% and 0.1% WSF diets enhanced antibody levels on days 14 and 21 (*p* < 0.05; Figure 8a). In HSF-fed fish, 0.01% and 0.1% doses significantly boosted antibody responses on most days, while the 1% group showed increases on days 7 and 14 (Figure 8b). With three weeks of feeding, 1% WSF significantly elevated antibody titres across all time points (Figure 9a). The 0.1% WSF group showed increased response on days 7 and 21, while 0.01% WSF did not produce any change. All HSF dose groups demonstrated significant antibody enhancement on nearly all tested days (Figure 9b).

### 3.3. Disease Resistance

Overall, feeding with *S. trilobatum* leaf extracts reduced fish mortality after bacterial challenge. In the WSF-fed groups, a significant reduction in mortality was seen only in the 1% group after one week (63.33%, RPS = 20.83) and the 0.01% group after three weeks (66.67%, RPS = 20) (Figure 10a; Table 1).

Three weeks of feeding with any HSF dose significantly reduced mortality (*p* < 0.05; Figure 10b; Table 1). Notably, the 0.01% and 0.1% HSF doses lowered mortality after just one week (RPS = 20.83 and 12.5, respectively). The 0.1% dose continued to offer protection after two weeks (36.67%, RPS = 54.17).

### 3.4. GC-MS Analysis

GC-MS profiling revealed 31 compounds in WSF and 44 in HSF. In the WSF, γ-sitosterol (RT: 27.424) had the largest peak area (1,959,571) and was identified as a major phytosterol. Other components included alkanes, fatty alcohols, quinones, diterpenes, triterpenes, and sesquiterpenoids (Table 2).

In HSF, the dominant compound was germacrene D-4-ol (RT: 11.962) with a peak area of 6,051,885, followed by abietinol. Table 3 lists additional constituents including diterpenes, fatty acids, esters, steroids, and other phytosterols.

### 3.5. In Silico Evaluation of Bioavailability of the Phytoconstituents

Based on Lipinski’s Rule of Five, one compound from the WSF and eleven from the HSF were predicted to have good oral bioavailability. The total relative abundance (sum of peak area percentages) for these compounds was 41.11% in WSF and 47.66% in HSF. Most violations of Lipinski’s criteria were associated with high miLogP values, suggesting limited solubility and absorption for those compounds.

## 4. Discussion

Medicinal plants are widely recognized for their ability to promote fish growth, stimulate appetite, and improve disease resistance by enhancing the immune system [27]. They can also contribute to antioxidant defence mechanisms, as shown in several species [28]. A recent meta-analysis by Mbokane and Moyo [29] supports the inclusion of medicinal plants in aquafeeds to strengthen disease resistance and improve overall fish health, which could have practical benefits for aquaculture operations. The current study reinforces these conclusions: enhanced immune responses were observed in *O. mossambicus* following dietary inclusion of *S. trilobatum* leaf extracts.

Serum globulin is a key biomarker of immune competence, elevated innate immune responses [30], and general health in fish [31]. In this study, oral administration of *S. trilobatum* fractions resulted in a significant elevation of serum globulin levels. These results are in line with previous reports showing increased globulin in juvenile greasy groupers (*Epinephelus tauvina*) when fed methanolic extracts of *Ocimum sanctum* or *Withania somnifera* [32].

Lysozyme, a vital component of fish innate immunity, was significantly upregulated in *O. mossambicus* receiving *S. trilobatum*-supplemented diets. Similar trends have been documented in *O. niloticus* fed with *Astragalus radix* root at 0.1–0.5% for one week [33], and in *Labeo rohita* supplemented with *Achyranthes aspera* extracts [31] or *Allium sativum* powder [34]. The observed enhancement of lysozyme activity may reflect increased macrophage numbers [35] or greater production of lysozyme per cell [36], as macrophages are the primary producers of this enzyme.

Antiprotease activity, another humoral defence mechanism, was also increased after feeding with the HSF fraction. Such proteins help neutralize bacterial proteases, limiting infection [37]. Comparable responses were observed in *Catla catla* and *Oncorhynchus mykiss* with dietary inclusion of *A. aspera* [38] or natural carotenoids in a study by Thompson et al. [39].

Enhanced ROS and RNI production in the treated groups suggests an activated oxidative burst response, a major defence mechanism of phagocytes [40]. The stimulation may be linked to increased leukocyte activity, as seen in *Glycyrrhiza glabra* [41] or *Eclipta alba*-treated fish [25] or due to the elevated phagocytic activity and cytokine [42]. However, species-specific responses are evident; for example, *Zingiber officinale* improved extracellular but not intracellular burst in trout [43].

MPO, a marker for neutrophil activation, was upregulated following dietary treatment. Similar enhanced activity has been observed in *C. carpio*, *Sparus aurata*, and *O. mykiss* fed with oak leaf or yeast [44,45,46]. The GC-MS results of *S. trilobatum* indicated the presence of carbohydrate-rich components, potentially responsible for the immune [25] activation observed, similar to β-glucan-based immunostimulants [36].

Immunostimulants may also improve specific immune functions, especially when followed by infection or vaccination [47]. Here, HSF-fed groups showed consistently higher antibody responses across multiple time points, while WSF also enhanced responses at select time points. This is consistent with earlier work involving *T. cordifolia* [48], *O. sanctum* [49], and Azadiractin [50].

Upon challenge with *A. hydrophila*, a significant reduction in mortality was observed in groups fed with both WSF and HSF, especially 0.1% HSF which produced the most consistent protective effect after 2 weeks. This agrees with earlier protective effects reported using *R. officinalis* [51] and *A. paniculata* [52]. Enhanced overall immunity has been reported upon dietary or IP administration of plant-derived immunostimulants in *O. mossambicus* [53,54], *L. rohita* [34], and *C. carpio* [55].

However, oral delivery offered slightly lower protection compared to intraperitoneal routes [9], possibly due to inconsistent ingestion, compound degradation in the digestive tract, or limited absorption [56].

The phytochemical composition of *S. trilobatum* likely contributed to these results. Compounds such as sobatum (β-sitosterol) are known to modulate T-helper cell responses and cytokine production in mammalian systems [57]. HSF, rich in di- and triterpenoids and saponins, showed superior bioactivity, where saponins are known to enhance cytokine production and lymphocyte proliferation [58]. Bioavailability is key when delivering plant-based compounds via feed. In this study, HSF contained more bioavailable components, including Germacrene d-4-ol, a lipophilic sesquiterpenoid present in high concentration and compliant with Lipinski’s Rule of Five. Similar compounds like nerolidol have shown protective effects in infected *N. tilapia* [59]. α– sitosterol extracted from *Streptomyces misakiensis* strain enhanced the growth, exhibited antioxidant properties and enhanced the immune status of the fish *O. niloticus* [60] and was found to be an effective anti-fungal agent. Similarly, feeding β-sitosterol-supplemented diet to large yellow croaker [61] and large mount bass [62] enhanced the intestinal immune function and survival against *A. hydrophila*. Similar sitosterols found to be in HSF (4.26 area%) and WSF (19.13%) might be responsible for the enhanced immune status in our study. Other compounds with poor solubility, indicated by high miLogP values, may have limited effectiveness due to poor absorption.

Taken together, both fractions of *S. trilobatum* stimulated disease resistance, specific immunity, and humoral and cellular nonspecific responses. However, certain doses of HSF outperformed WSF in elevating RNI production after 1 week and PBL MPO activity after 3 weeks of feeding. Additionally, HSF exhibited a significant difference in elevating antibody response after 1 week of feeding on the 14th day indicating a more robust and rapid adaptive immune activation. This corresponds with a notable reduction in the percentage mortality in the group fed with 0.1% HSF-supplemented diet for 2 weeks and was consistent till 3 weeks, showing higher efficacy. This suggests that 0.1% HSF could be explored as a feed-based prophylactic immunostimulant in aquaculture systems. These findings also suggest that prolonged continuous supplementation of immunostimulant does not necessarily improve outcomes and may place additional metabolic burden on the fish as well as unnecessary cost on the farmer.

## 5. Conclusions

Non-specific immune parameters, antibody response, and resistance to bacterial challenge in *Oreochromis mossambicus* were evaluated after the administration of either water-soluble fraction (WSF) or hexane-soluble fraction (HSF) of *Solanum trilobatum* leaves as feed supplement for 1, 2, or 3 weeks. Both the fractions increased serum globulin levels, lysozyme, and antiprotease activity, and in the peripheral blood leukocytes MPO content and ROS production also showed an elevated response. The antibody response was significantly higher in the HSF-fed group, and this reflected the reduced mortality in this group, whereas WSF could reduce the mortality only after 1 or 3 weeks. We found a better performance of HSF in stimulating immunity, which might be due to the presence and bioavailability of aromatic compounds and phytosterols when compared to low molecular weight alcohols and carbonyls in WSF. These findings demonstrate that HSF enhances both innate and adaptive immunity in tilapia and can be explored for administration through feed to enhance the overall immunity of fish in aquaculture.

## Figures and Tables

**Figure 1 biology-14-01333-f001:**
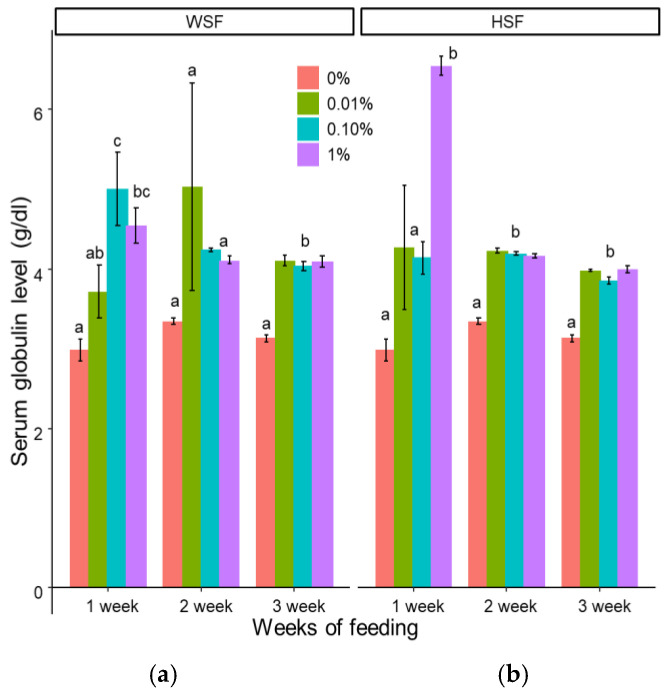
Modulation of serum globulin level in fish fed with WSF (**a**) or HSF (**b**) of *S. trilobatum* leaf-supplemented diet for 1, 2, and 3 weeks. The levels of significance were expressed as *p*-values less than 0.05, indicating significant differences within a week, as provided in the graph with different letters.

**Figure 2 biology-14-01333-f002:**
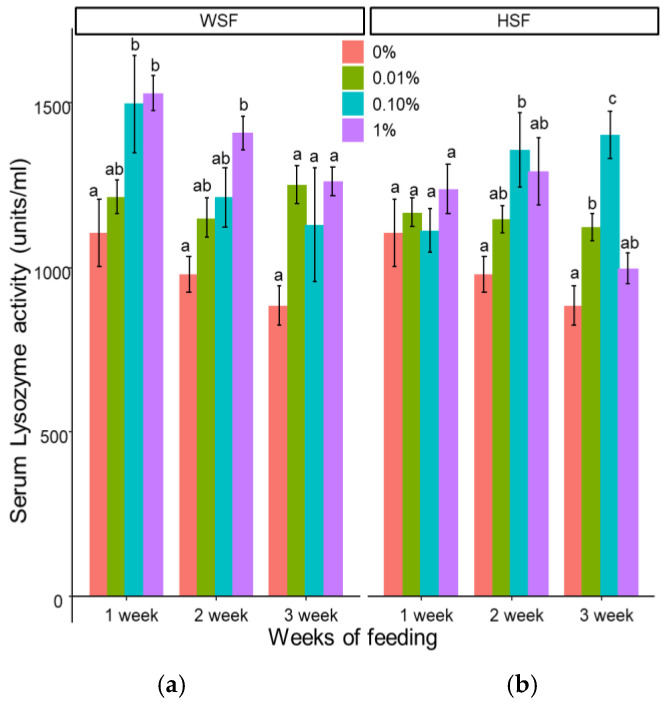
Modulation of serum lysozyme activity in fish fed with WSF (**a**) or HSF (**b**) of *S. trilobatum* leaf-supplemented diet for 1, 2, and 3 weeks. The levels of significance were expressed as *p*-values less than 0.05, indicating significant differences within a week, as provided in the graph with different letters.

**Figure 3 biology-14-01333-f003:**
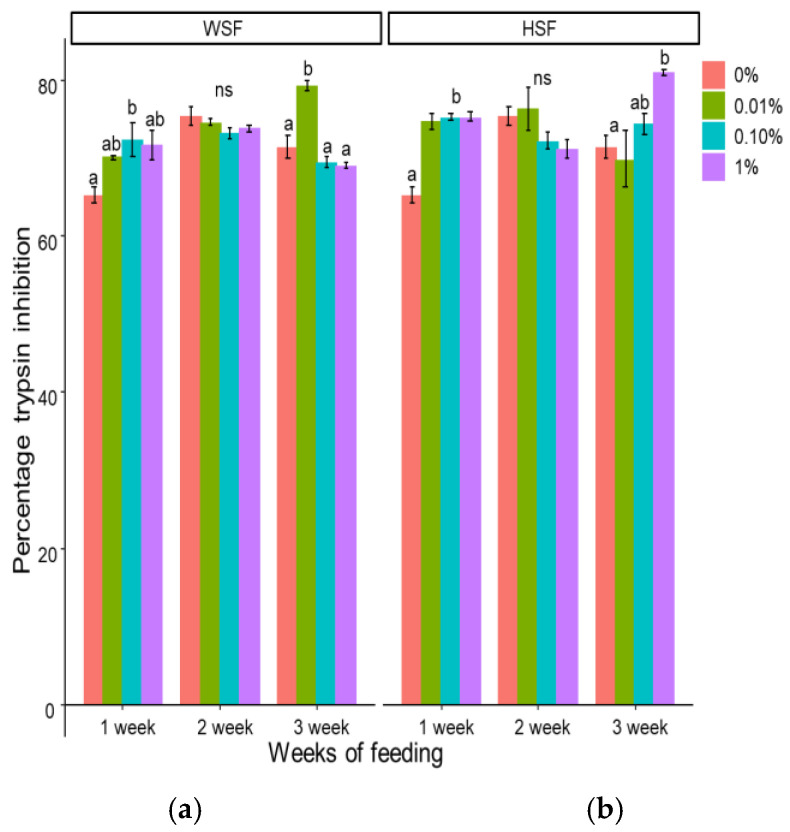
Modulation of serum antiprotease activity in fish fed with WSF (**a**) or HSF (**b**) of *S. trilobatum* leaf-supplemented diet for 1, 2, and 3 weeks. The levels of significance were expressed as *p*-values less than 0.05, indicating significant differences within a week, as provided in the graph with different letters. ns: not significant.

**Figure 4 biology-14-01333-f004:**
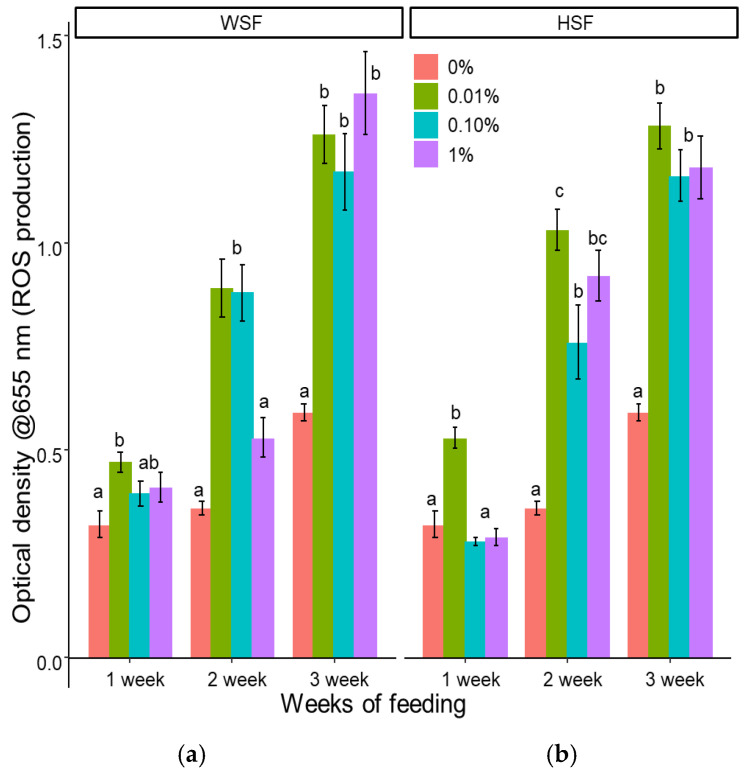
Modulation of ROS production in fish fed with WSF (**a**) or HSF (**b**) of *S. trilobatum* leaf-supplemented diet for 1, 2, and 3 weeks. The levels of significance were expressed as *p*-values less than 0.05, indicating significant differences within a week, as provided in the graph with different letters.

**Figure 5 biology-14-01333-f005:**
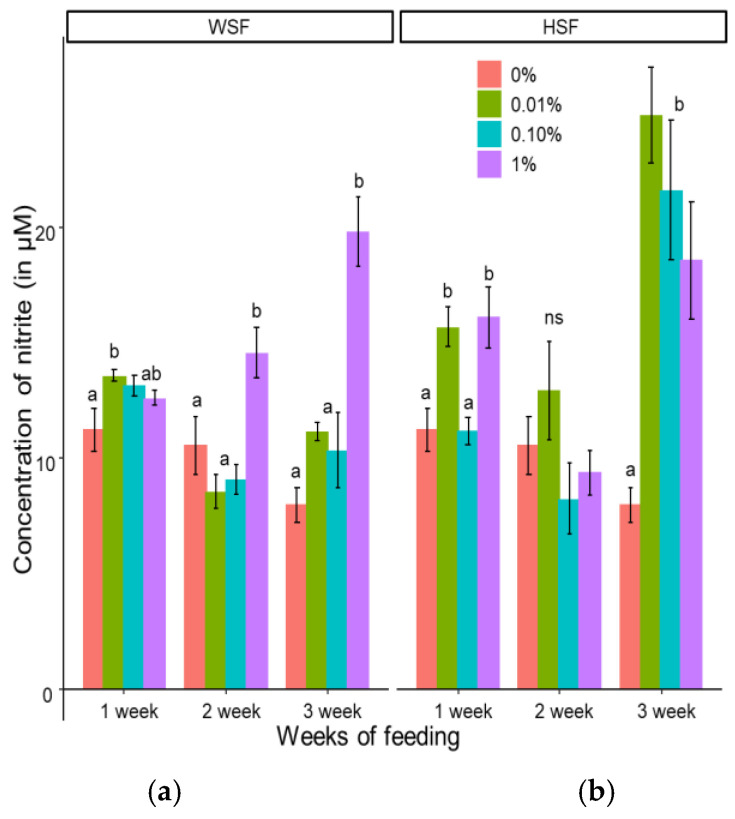
Modulation of RNI production in fish fed with WSF (**a**) or HSF (**b**) of *S. trilobatum* leaf-supplemented diet for 1, 2, and 3 weeks. The levels of significance were expressed as *p*-values less than 0.05, indicating significant differences within a week, as provided in the graph with different letters. ns: not significant.

**Figure 6 biology-14-01333-f006:**
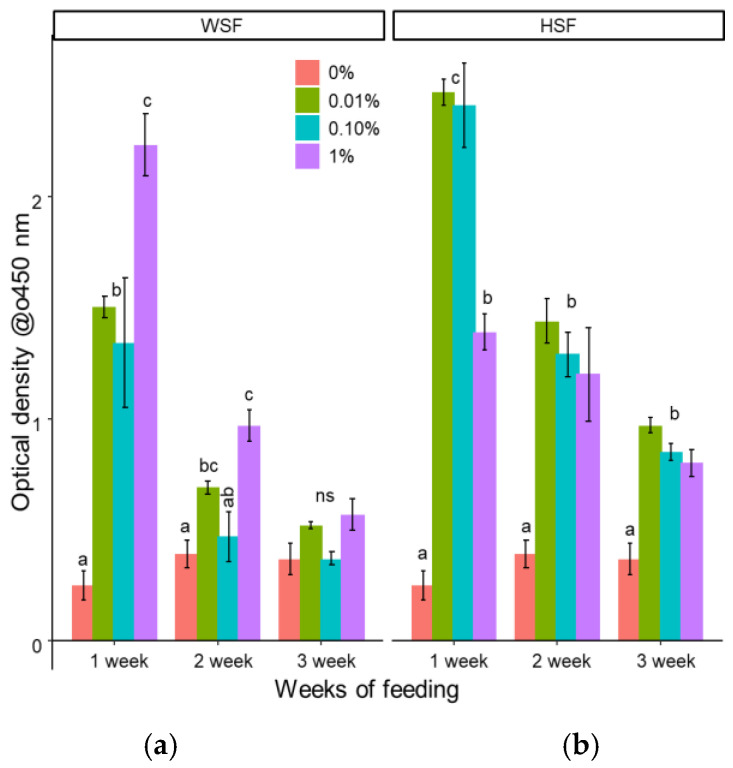
Modulation of MPO content in fish fed with WSF (**a**) or HSF (**b**) of *S. trilobatum* leaf-supplemented diet for 1, 2, and 3 weeks. The levels of significance were expressed as *p*-values less than 0.05, indicating significant differences within a week, as provided in the graph with different letters. ns: not significant.

**Figure 7 biology-14-01333-f007:**
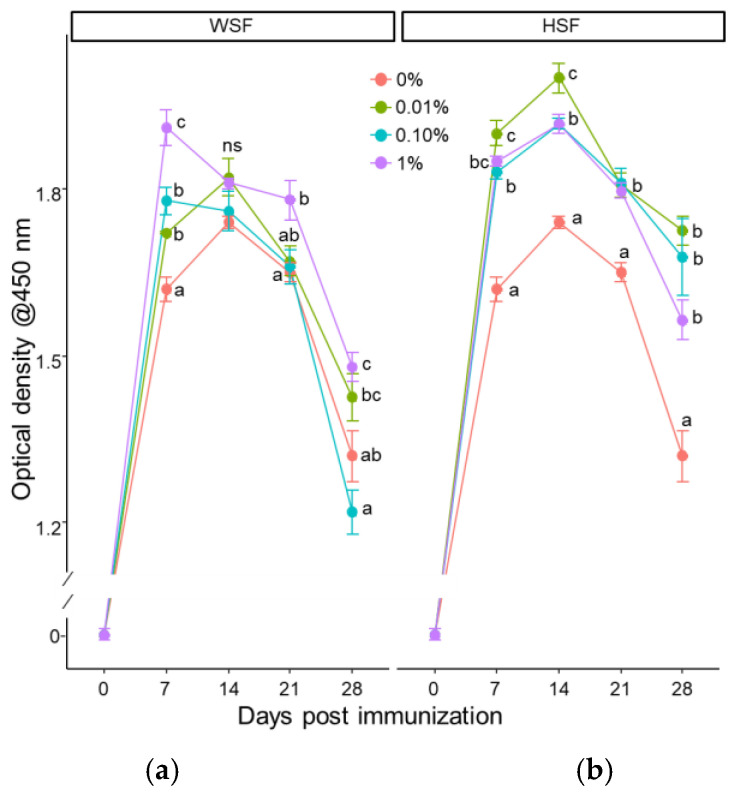
Modulation of antibody response to heat-killed *A. hydrophila* tested by ELISA in fish fed with WSF (**a**) or HSF (**b**) of *S. trilobatum* leaf-supplemented diet for 1 week. The levels of significance were expressed as *p*-values less than 0.05, indicating significant differences within a week, as provided in the graph with different letters. ns: not significant.

**Figure 8 biology-14-01333-f008:**
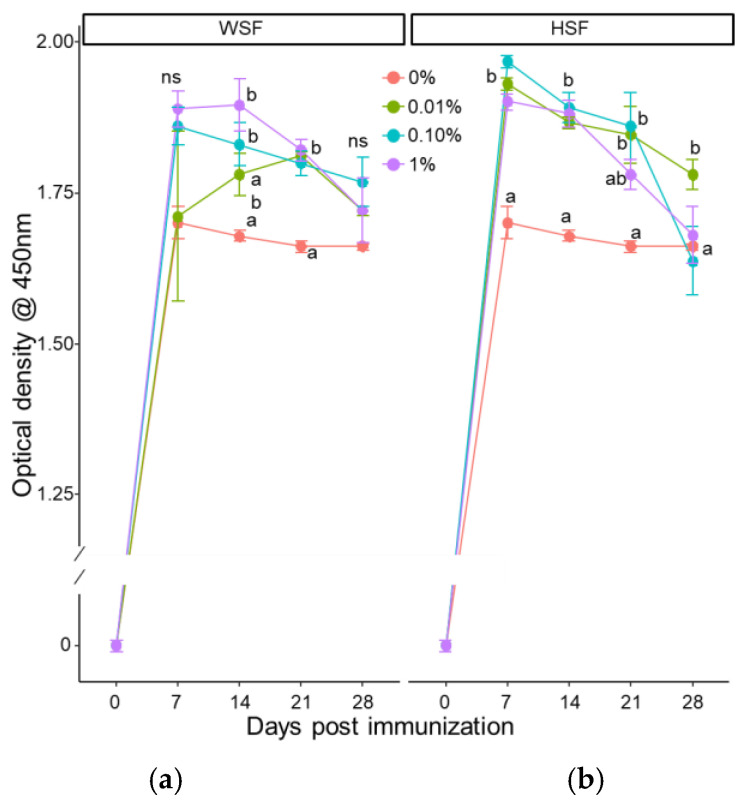
Modulation of antibody response to heat-killed *A. hydrophila* tested by ELISA in fish fed with WSF (**a**) or HSF (**b**) of *S. trilobatum* leaf-supplemented diet for 2 weeks. The levels of significance were expressed as *p*-values less than 0.05, indicating significant differences within a week, as provided in the graph with different letters. ns: not significant.

**Figure 9 biology-14-01333-f009:**
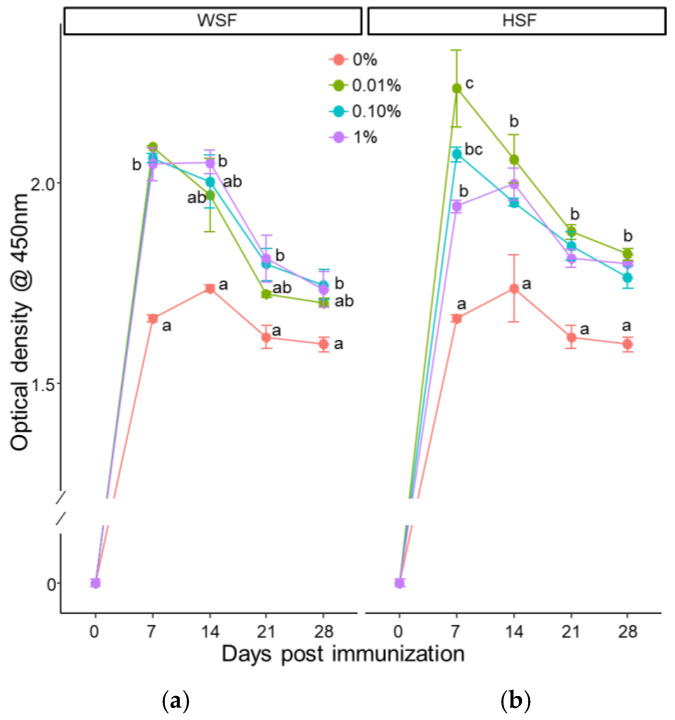
Modulation of antibody response to heat-killed *A. hydrophila* tested by ELISA in fish fed with WSF (**a**) or HSF (**b**) of *S. trilobatum* leaf-supplemented diet for 3 weeks. The levels of significance were expressed as *p*-values less than 0.05, indicating significant differences within a week, as provided in the graph with different letters.

**Figure 10 biology-14-01333-f010:**
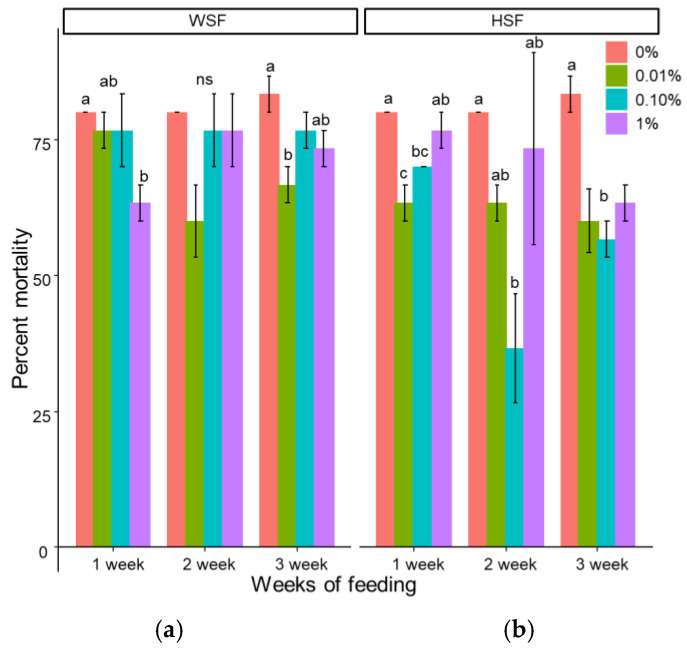
Modulation of disease resistance in fish fed with WSF (**a**) or HSF (**b**) of *S. trilobatum* leaf-supplemented diet for 1, 2, and 3 weeks. The levels of significance were expressed as *p*-values less than 0.05, indicating significant differences within a week, as provided in the graph with different letters. ns: not significant.

**Table 1 biology-14-01333-t001:** Change in relative percent survival (RPS) in fish fed with WSF or HSF of *S. trilobatum* leaf-supplemented diet for 1, 2, and 3 weeks.

Fractions	Dose	1 Week	2 Weeks	3 Weeks
WSF	0.01%	4.17	25.00	20.00
	0.1%	4.17	4.17	8.00
	1%	20.83	4.17	12.00
HSF	0.01%	20.83	20.83	28.00
	0.1%	12.50	54.17	32.00
	1%	4.17	8.33	24.00

**Table 2 biology-14-01333-t002:** Phytoconstituents in *S. trilobatum* WSF based on GC-MS analysis and the number of Lipinski’s violations.

Chemical Class	Retention Time	Area	Area %	Compound Name	Molecular Formula	Molecular Weight, g/mol	N Violations
Alcohol	22.207	103,451	0.59	1,3-Propanediol, dodecyl ethyl ether	C_17_H_36_O_2_	272	1
Alkanes	12.338	703,424	4.05	Nonane, 3-methyl-5-propyl-	C_13_H_28_	184	1
	19.053	188,710	1.09	2-methylhexacosane	C_27_H_56_	380	1
	14.097	118,161	0.68	Heptadecane, 3-methyl-	C_18_H_38_	254	1
Alkylbenzene	15.765	636,438	3.66	Benzenepropanoic acid, 3,5-bis(1,1-dimethylethyl)-4-hydro	C_18_H_28_O_3_	292	1
Benzenoids	7.449	6,997,607	40.24	Azulene	C_10_H_8_	128	0
Carbohydrate	18.189	48,125	0.28	Carbonic acid, eicosyl vinyl ester	C_23_H_44_O_3_	368	1
Diterpene	17.473	273,775	1.57	Phytol	C_20_H_40_O	296	1
	14.793	127,124	0.73	Neophytadiene	C_20_H_38_	278	1
	18.342	66,102	0.38	Phytol, acetate	C_22_H_42_O_2_	338	1
Ergostane steroids	26.509	684,277	3.94	Ergost-5-en-3-ol, (3.β.,24r)-	C_28_H_48_O	400	1
Fatty acid methyl ester	22.462	181,667	1.04	Tricosanoic acid, methyl ester	C_24_H_48_O_2_	368	1
Fatty Alcohol	20.733	180,506	1.04	1-hexacosanol	C_26_H_54_O	382	1
	17.3	161,875	0.93	6,11-hexadecadien-1-ol	C_16_H_30_O	238	1
Fattyacids esters	17.349	155,442	0.89	Methyl Stearate	C_19_H_36_O_2_	296	1
	17.574	67,628	0.39	Octadecanoic acid, methyl ester	C_19_H_38_O_2_	298	1
Hydrocarbon	10.196	1,247,832	7.18	1-chlorohexadecane	C_16_H_33_Cl	260	1
Long chain fatty alcohol	30.253	162,922	0.94	1,2-nonadecanediol	C_19_H_40_O_2_	300	1
Macrocyclin diterpene alcohol	29.191	196,398	1.13	Thunbergol	C_20_H_34_O	290	1
Pentracyclic triterpenoid	28.395	429,331	2.47	9,19-Cyclolanost-24-en-3-ol, (3.β.)-	C_30_H_50_O	426	1
Phytosterol	27.424	1,959,571	11.27	γ-sitosterol	C_29_H_50_O	414	1
	26.749	1,210,014	6.96	Stigmasta-5,23-dien-3-ol, (3.β.)-	C_29_H_48_O	412	1
	25.533	156,360	0.9	Cholest-5-en-3-ol (3.β)-	C_27_H_46_O	386	1
Quinone and hydroquinone lipids	25.436	151,482	0.87	Vitamin E	C_29_H_50_O_2_	430	1
Sesquiterpenoids	16.382	79,887	0.46	Heptadecane, 2,6,10,15-tetramethyl-	C_21_H_44_	296	1
Sesterterpenoids	23.279	67,717	0.39	α-tocospiro b	C_29_H_50_O_4_	462	1
Triterpene	23.039	256,270	1.47	Squalene	C_30_H_50_	410	1
	25.225	325,188	1.87	24-Norursa-3,12-diene	C_29_H_46_	394	1

**Table 3 biology-14-01333-t003:** Phytoconstituents in *S. trilobatum* HSF based on GC-MS analysis and the number of Lipinski’s violations.

**Chemical Class**	**Retention Time**	**Area**	**Area %**	**Compound Name**	**Molecular Formula**	**Molecular Weight, g/mol**	**N Violations**
Carboxylic ester	8.134	83,253	0.47	4-tert-butylcyclohexyl acetate	C_12_H_22_O_2_	198	0
Diterpene	14.715	156,057	0.89	Neophytadiene	C_20_H_38_	278	1
	15.822	113,913	0.65	Biformene	C_20_H_32_	272	1
	17.387	1,146,616	6.54	Abieta-7,13-diene	C_20_H_32_	272	1
	17.823	192,602	1.10	Verticiol	C_20_H_34_O	290	1
	17.888	656,202	3.74	Agathadiol	C_20_H_34_O_2_	306	0
	17.972	93,879	0.54	Neoabietadiene	C_20_H_32_	272	1
	18.375	156,065	0.89	Isodextropimaraldehyde	C_20_H_30_O	286	1
	18.630	64,779	0.37	Palustrinal	C_20_H_30_O	286	1
	18.755	172,913	0.99	Levopimarate	C_21_H_32_O_2_	316	1
	18.953	86,383	0.49	Abieta-8,11,13-trien-18-a	C_20_H_28_O	284	1
	19.317	1,296,637	7.39	Abietinol	C_20_H_32_O	288	1
	19.924	113,206	0.65	Neo abietal	C_20_H_30_O	286	1
Ergostane steroids	26.549	85,471	0.49	Ergost-5-en-3-ol, (3.β.,24r)-	C_28_H_48_O	400	1
Fatty acids	22.477	212,007	1.21	Galaxolide	C_14_H_26_O_4_	258	0
	16.674	166,324	0.95	Tetradecanedioic acid, 3-oxo-, dimethyl ester	C_16_H_28_O_5_	300	0
Fatty acid ester	17.334	162,807	0.93	9-octadecenoic acid, methyl ester	C_19_H_36_O_2_	296	1
	17.562	152,150	0.87	Methyl stearate	C_19_H_38_O_2_	298	1
	15.693	1,029,561	5.87	Hexadecanoic acid, methyl ester	C_17_H_34_O_2_	270	1
Fatty alcohol	14.820	30,242	0.17	1-tetradecanol	C_14_H_30_O	214	1
Organic hetero tricyclic	14.883	279,538	1.59	Hexamethyl-pyranoindane	C_18_H_26_O	258	1
Phytosterol	25.240	256,252	1.46	Stigmast-5-en-3-ol, (3.β.)-	C_29_H_50_O	414	1
	26.801	190,485	1.09	Stigmasterol	C_29_H_48_O	412	1
	27.474	299,140	1.71	γ-sitosterol	C_29_H_50_O	414	1
Polycyclic aromatic hydrocarbons	7.360	378,839	2.16	Azulene	C_10_H_8_	128	0
PUFA	17.288	128,833	0.73	Verticillol	C_18_H_32_O_2_	280	1
Sesquiterpene	9.534	170,825	0.97	Β -elemene	C_15_H_24_	204	1
	8.776	100,393	0.57	Cyclohexene, 4-ethenyl-4-methyl-3-(1-methylethenyl)-1-(1	C_15_H_24_	204	1
	9.729	140,414	0.80	Cyperene	C_15_H_24_	204	0
	10.053	580,731	3.31	Germacrene b	C_15_H_24_	204	1
	10.767	217,949	1.24	Germacrene d	C_15_H_24_	204	1
	10.974	488,025	2.78	Cedrelanol	C_15_H_26_O	222	0
	11.263	41,022	0.23	Cubebol	C_15_H_26_O	222	0
	11.962	6,051,885	34.51	Germacrene d-4-ol	C_15_H_26_O	222	0
	13.393	347,188	1.98	Shyobunol	C_15_H_26_O	222	1
	14.253	86,532	0.49	Abietinal	C_15_H_26_O	222	0
	18.557	55,406	0.32	10,11-dihydroxy-3,7,11-trimethyl-2,6-dodecadienyl acetate	C_17_H_30_O_4_	298	0
Steroid	13.979	599,432	3.42	Ergostane-5,25-diol	C_39_H_76_O_6_Si_3_	724	2
Tetralins	14.990	191,432	1.09	Tonalid	C_18_H_26_O	258	
Triterpenoid	16.499	136,199	0.78	Manool oxide	C_20_H_34_O	290	1
	14.190	78,086	0.45	7-dimethyl(chloromethyl)silyloxytridecane	C_16_H_35_ClOSi	306	1
	21.061	323,290	1.84	Bis(2-ethylhexyl) phthalate	C_24_H_38_O_4_	390	

## Data Availability

Data available with corresponding author will be provided upon request.

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
