# Peer review of "Differential Immunostimulatory Effects of Hydrophilic and Hydrophobic Solanum trilobatum Fractions in Tilapia"

_biology, 2025, doi:10.3390/biology14101333_

Round 1

Reviewer 1 Report

Comments and Suggestions for Authors

In the current manuscript, the authors examined the potential effects of the Solanum trilobatum leaf extracts on both non-specific and specific immune responses in tilapia. Additionally, a challenge with Aeromonas hydrophila demonstrated reduced mortality rates in fish administered these extracts. These findings underscore the potential of Solanum trilobatum leaf extracts as an immunomodulatory agent for tilapia in aquaculture. However, several concerns need addressing, as outlined below.

  1. Page 3: The body length and weight of thetilapia used in this study, as well as the water temperature should be documented. The number of fish in each experimental group should be specified.
  2. Page 4: The methodfor detecting superoxide anion and nitric oxide production by peripheral blood leukocytes needs clarification. Were the peripheral blood leukocytes isolated for this purpose, or was the content of superoxide anions and nitric oxide measured directly in the serum?
  3. Page 4: It is recommended to assess the bacterial abundance in the liverof
  4. Growth performance is a crucial indicator for evaluating the effectsof immunomodulators.Therefore, the growth parameters should be detailed.
  5. This studyanalyzed the effects of different extraction methods for Solanum trilobatum leaves, and dosages and application durations of two extracts, on the non-specific and specific immune responses and disease resistance of tilapia. It is recommended that the most optimal usage be determined in the discussion or conclusion section. 

Author Response

In the current manuscript, the authors examined the potential effects of the Solanum trilobatum leaf extracts on both non-specific and specific immune responses in tilapia. Additionally, a challenge with Aeromonas hydrophila demonstrated reduced mortality rates in fish administered these extracts. These findings underscore the potential of Solanum trilobatum leaf extracts as an immunomodulatory agent for tilapia in aquaculture. However, several concerns need addressing, as outlined below.

Thank you very much for taking the time to review this manuscript. Please find the detailed responses below and the corresponding revisions/corrections in track changes in the re-submitted files.

Yes

Can be improved

Must be improved

Not applicable

Response

Does the introduction provide sufficient background and include all relevant references?

(x)

( )

( )

( )

Is the research design appropriate?

(x)

( )

( )

( )

We have revised the methodology section according to the reviewer’s comment.

Are the methods adequately described?

( )

(x)

( )

( )

Are the results clearly presented?

(x)

( )

( )

( )

Are the conclusions supported by the results?

(x)

( )

( )

( )

Are all figures and tables clear and well-presented?

(x)

( )

( )

( )

Comment 1: Page 3: The body length and weight of the tilapia used in this study, as well as the water temperature should be documented. The number of fish in each experimental group should be specified.

The weight of the fish was 25 ± 5 g for all assays, except for the nonspecific cellular assays, where 50 ± 5 g fish were used to obtain sufficient peripheral blood leukocytes. The corresponding lengths were 11.63 ± 0.12 cm and 14.29 ± 0.21 cm, respectively. The water temperature was maintained at 28 ± 2 °C. This information has been added to the revised manuscript (lines 111–120). The number of fish used in each experiment is already indicated in lines 128, 168, and 180.

Comment 2: The method for detecting superoxide anion and nitric oxide production by peripheral blood leukocytes needs clarification. Were the peripheral blood leukocytes isolated for this purpose, or was the content of superoxide anions and nitric oxide measured directly in the serum?

Peripheral blood leukocytes were isolated using Lymphosep (ICN biomedicals, USA) density gradient. After viable cell counting, the assays were carried out. This information has been added to the revised manuscript (lines 132 and 134).

Comment 3: Page 4: It is recommended to assess the bacterial abundance in the liver of

Thank you for this recommendation. We will include this aspect in future studies.

Comment 4: Growth performance is a crucial indicator for evaluating the effects of immunomodulators. Therefore, the growth parameters should be detailed.

Given that the primary objective of this study was to evaluate the immunomodulatory effects of Solanum trilobatum extracts, growth parameters were not formally assessed. While this represents a limitation of the current work, we plan to incorporate comprehensive growth performance metrics—including weight gain, specific growth rate, and feed conversion ratio—in future investigations to provide a more holistic assessment of the extract’s potential in aquaculture applications.

Comment 5: This study analyzed the effects of different extraction methods for Solanum trilobatum leaves, and dosages and application durations of two extracts, on the non-specific and specific immune responses and disease resistance of tilapia. It is recommended that the most optimal usage be determined in the discussion or conclusion section. 

The suggested information has been added to the revised manuscript (lines 426 and 428).

Reviewer 2 Report

Comments and Suggestions for Authors

Congratulations on this manuscript. It is recommended to improve the methodology in the sample treatment part and indicate the equipment and reagents used in the research.

Author Response

Congratulations on this manuscript. It is recommended to improve the methodology in the sample treatment part and indicate the equipment and reagents used in the research.

Many thanks for the encouragement. We have updated the equipment and reagents used in the present study.

  Yes Can be improved Must be improved Not applicable Response
Does the introduction provide sufficient background and include all relevant references? ( ) (x) ( ) ( ) We have rewritten the introduction section
Is the research design appropriate? ( ) (x) ( ) ( ) We are following this research design in many of our published studies. We are open to reviewer's comment regarding future improvements.
Are the methods adequately described? ( ) (x) ( ) ( ) The methods section has been updated.
Are the results clearly presented? (x) ( ) ( ) ( )  
Are the conclusions supported by the results? (x) ( ) ( ) ( )  
Are all figures and tables clear and well-presented? ( ) (x) ( ) ( ) We have improved the figures to better present our results.

Comment 1: Could you explain to me why bioavailability represents an important limitation in the effectiveness of some natural products?

Response 1: The effectiveness of the water-soluble fraction (WSF) may be limited by bioavailability. Many polar compounds in WSF exhibit low membrane permeability and may undergo rapid metabolism or elimination, reducing the fraction of active compounds that reach target physiological sites. These factors can limit the in vivo efficacy of the extract by limiting the bioactive compound reaching immune cells. In contrast, the hexane-soluble fraction (HSF) contains many nonpolar compounds with more drug-like properties, which may be more readily absorbed by the immune cells.

Comment 2: Please describe in more detail the processing of Solanum leaves as the description does not clearly show how the fraction used in the food was selected or how the leaf was used directly?

Response 2: Preparation methods for the water-soluble fraction (WSF) and hexane-soluble fraction (HSF) followed the published protocol by Divyagnaneswari et al. (2007).
The doses selected for feed administration were based on earlier studies in which S. trilobatum extracts were administered intraperitoneally. Following careful dose conversion and safety considerations, the final concentrations used in this study were chosen to achieve immunomodulatory effects analogous to those observed in prior injectable protocols Divyagnaneswari et al. (2007).

Comment 3: It would be good to mention whether the identified compounds have been evaluated in similar models and also indicate if there are compounds that are present in WSF and HSF, also indicate their abundance in the sample?

Response 3: A similar compound has been evaluated in a related model, as referenced in [59]. There are few reports on the phytosterols which have been added now in the discussion (line number: 411-416). 
 The GC–MS analysis table lists the compounds with the highest peak area percentages, which reflect their relative abundance in the sample. Compounds present in the WSF and HSF are indicated in the table.

Comment 4: How does the effect obtained with S. trilobatum on O. mossambicus compare with other studies on different fish species?

Response 4: The modulation of immune parameters in other fish species by various plant extract is discussed in the ‘Discussion’ section. In our previous studies with various plant and algal extracts, we observed similar patterns: mid doses administered over two weeks elicited the maximum immune response, which aligns with the findings in the present study.

Comment 5: Should some of the reported compounds that are used with effects associated with those evaluated in the study be mentioned?

Response 5: The present study reports the immunostimulatory properties of two different extracts from S. trilobatum in tilapia. Since crude extracts were used, it is not possible to attribute the observed effects to specific identified compounds. Therefore, any mechanistic links to individual compounds are presented as hypotheses rather than confirmed effects.

Comment 6: The HSF fraction may be a promising alternative for strengthening immunity in farmed fish. What prospects could these results raise?

Response 6: As mentioned in the ‘Conclusion’, the HSF fraction could be explored for application in aquaculture to prevent disease outbreaks and reduce mortality. Specific compounds can be isolated for further testing to identify precise immunomodulatory pathways, optimize dosage, and assess long-term safety.

Reviewer 3 Report

Comments and Suggestions for Authors

A brief summary:

This MS by Divya Gnaneswari et al. fed tilapia either water (WSF) or hexane soluble fraction (HSF) of Solanum trilobatum leaves for 1, 2, or 3 weeks to assess their effects on nonspecific immune responses, antibody response, and disease resistance to bacterial challenge after each feeding period. Results showed that both extracts improved the immune system over time, by enhancing globulin level, lysozyme, and anti-protease activity. HSF enhanced key immune responses like ROS and MPO and led to higher antibody levels than WSF.

  • General concept comments.

Overall, I thought this was a study in a system with limited previous knowledge of this level on this specific topic. I appreciated their multi-faceted approach and the time course involved in this work and think it add significant merit to their work. I do think that their method was clear, and the overall conclusions were related to results. I think this is good work, but should undergo major revision before acceptance.

  • Specific comments

Line 68  The sentence is missing a punctuation mark at the end.

Line 68  When citing the author's name in the text, the authors should make correct citations according to the journal requirements.

“Subramani et al., [4]” change to “Subramani et al. [4]”

Line 75-76 The author should provide the proof of their research sources, such as published references.

Line 80 The sentence is missing a punctuation mark at the end.

Line 109 what about the size (Body length and weight) of male Oreochromis mossambicus?

Line 112-113  The author should refer to the journal format, and data do not need to be enclosed in parentheses. “(5.2 ± 0.1 mg/L)” change to “5.2 ± 0.1 mg/L’

Line 113 How to monitor the pH and dissolved oxygen? make it clear.

Line 125 How to obtain blood samples? make it clear.

Line 126-127 What is the previously published methods. Was it the same as in Ref. 9. Provide a brief description.

Line 133-134,182, …361-362  When citing the author's name in the text, the authors should make correct citations according to the journal requirements.  

Line 143-144 “microplate reader” Provide model and producer information.

Line 210 Has the author conducted any analysis on the normal distribution of the data?

Line 236-237,247-248, 259-260, 269-270,278-279,286-287,296-297,299-300  The sentence repeated in Figure 1. These are all the same sentences. The author can provide a unified explanation in the data analysis, rather than explaining each sentence for each figure.

Finally, I suggest the author should carefully review and revise the entire text again. Some are missing punctuation marks, which could have been avoided.

Author Response

Quality of English Language (x) The English could be improved to more clearly express the research.
( ) The English is fine and does not require any improvement.   The English language has been thoroughly revised with the assistance of ChatGPT to improve clarity and readability. We believe the revised version now communicates the research clearly and effectively.   Comments and Suggestions for Authors

A brief summary:

This MS by Divya Gnaneswari et al. fed tilapia either water (WSF) or hexane soluble fraction (HSF) of Solanum trilobatum leaves for 1, 2, or 3 weeks to assess their effects on nonspecific immune responses, antibody response, and disease resistance to bacterial challenge after each feeding period. Results showed that both extracts improved the immune system over time, by enhancing globulin level, lysozyme, and anti-protease activity. HSF enhanced key immune responses like ROS and MPO and led to higher antibody levels than WSF.

  • General concept comments.

Overall, I thought this was a study in a system with limited previous knowledge of this level on this specific topic. I appreciated their multi-faceted approach and the time course involved in this work and think it add significant merit to their work. I do think that their method was clear, and the overall conclusions were related to results. I think this is good work, but should undergo major revision before acceptance.

We thank the reviewer for the encouraging comments. The manuscript has been thoroughly revised in line with the reviewers’ suggestions, and we believe the current version fully addresses the concerns raised and is suitable for publication.

  • Specific comments

Line 68  The sentence is missing a punctuation mark at the end.

The missing punctuation mark has been added.

Line 68  When citing the author's name in the text, the authors should make correct citations according to the journal requirements.

“Subramani et al., [4]” change to “Subramani et al. [4]”

The suggested format change has been carried out.

Line 75-76 The author should provide the proof of their research sources, such as published references.

It is given at the end of the paragraph (reference number 9).

Line 80 The sentence is missing a punctuation mark at the end.

The missing punctuation mark has been added.

Line 109 what about the size (Body length and weight) of male Oreochromis mossambicus?

Information regarding body length and weight of the fish has been added (lines 113-114).

Line 112-113  The author should refer to the journal format, and data do not need to be enclosed in parentheses. “(5.2 ± 0.1 mg/L)” change to “5.2 ± 0.1 mg/L’

We thank the reviewer for the helpful suggestion. The values have been revised to follow the journal format: “pH was 7.3 ± 0.3 and dissolved oxygen was 5.2 ± 0.1 mg/L,” without enclosing the data in parentheses (lines 116 and 117).

Line 113 How to monitor the pH and dissolved oxygen? make it clear.

We used handheld pH meter to measure the pH, and dissolved oxygen was measured using Winkler’s titration method. This has been updated in the manuscript (lines 117-118).

Line 125 How to obtain blood samples? make it clear.

The blood samples were collected from the common cardinal vein as mentioned in line no 129.

Line 126-127 What is the previously published methods. Was it the same as in Ref. 9. Provide a brief description.

We confirm that the same protocol for cell separation described in Ref. 9 was followed. As the 'Lymphosep' density gradient method is a well-established standard, we considered a detailed description unnecessary.

Line 133-134,182, …361-362  When citing the author's name in the text, the authors should make correct citations according to the journal requirements.  

The necessary changes have been made.

Line 143-144 “microplate reader” Provide model and producer information.

Microplate reader used in our study is Bio-Rad, USA. The information has been added in the revised manuscript (line 152).

Line 210 Has the author conducted any analysis on the normal distribution of the data?

Yes, prior to conducting ANOVA, the normal distribution of the data was assessed using the Shapiro–Wilk test, and homogeneity of variances was evaluated using Bartlett’s test. Both tests were performed automatically in SigmaStat. This information is now added to the manuscript (lines 219-225).

Line 236-237,247-248, 259-260, 269-270,278-279,286-287,296-297,299-300  The sentence repeated in Figure 1. These are all the same sentences. The author can provide a unified explanation in the data analysis, rather than explaining each sentence for each figure.

The suggested corrections have been carried out.

Finally, I suggest the author should carefully review and revise the entire text again. Some are missing punctuation marks, which could have been avoided.

The manuscript has been thoroughly revised for punctuation, grammar, and overall clarity to ensure it meets the journal’s standards.

Reviewer 4 Report

Comments and Suggestions for Authors

The work presented here is an important practical aspect of fish farming, ensuring protection against microbial infections. The authors describe the beneficial effects of using extracts from the Solanaceae family as protective agents for fish. Given the growing antibiotic resistance of bacteria, research on the properties of other antibacterial agents is important in fish farming and human health.
However, I have a few comments for the authors regarding the work presented.

1) Regarding point 2.4. How long was the monitoring period for antibody levels after exposing the fish to the extracts?

2) Regarding point 3.1.1: What differences were found in globulin levels after two weeks of HSF and WSF administration? Were there any significant differences between these groups? At what doses were these differences observed?

3) Due to the small group of fish tested, how many repetitions of measurements were performed in each analysis? Were the results averaged from only six readings?

4) Regarding point 3.1.2, did WSF extracts increase lysozyme activity compared to HSF? At what concentration and time period were the greatest differences found? 

5) Point 3.1.3. Please provide information on whether there were significant differences in anti-protease activity between WSF and HSF. Which compound improved this activity more effectively?

6) Regarding point 3.1.4: What is the reason for the large increase in values in graphs 4a and 4b after two and three weeks compared to one week? Was this sharp increase associated with a significant increase in ROS at specific doses between weeks? Were there any significant differences in the ROS levels generated by WSF and HSF in individual weeks and doses when compared directly?
6) Regarding point 3.1.5, what caused the decrease in MPO activity after two and three weeks for both solutions?

7) In the captions accompanying the graphs, the authors state that "errors with different letters indicate significant differences." It should be clarified whether this refers to bars within a week or between weeks, as this is sometimes unclear.

8) In Figures 7, 8, and 9, the letter markings indicating significance are illegible because the lines overlap too much and obscure the markings.

9) Why was the observation period for the immune response extended to four weeks when the other analyses were conducted over three weeks?

10) Lines 384-386: To what extent do differences in the level of protection depend on the method of administering plant extracts? Also, consider this study as an example.

11) Did the authors check how long the protective properties beneficial to the body persist after discontinuing feed with WSF and HSF additives? For example, resistance to disease? Or, must continuous supplementation be maintained for these properties to occur?

12) Did the authors perform microbiological analyses of the WSF and HSF extracts at the analyzed concentrations to confirm their bactericidal effect?

In general, the study lacks direct comparisons of the studied extracts. In the graphs, the authors refer to each extract separately, both in relation to the control and within concentrations. This is why I have questions about the direct differences between WSF and HSF; this is the only way to determine which is better, and the authors draw such conclusions.

Author Response

  Yes Can be improved Must be improved Not applicable Response
Does the introduction provide sufficient background and include all relevant references? (x) ( ) ( ) ( )  
Is the research design appropriate? (x) ( ) ( ) ( )  
Are the methods adequately described? ( ) (x) ( ) ( ) The methods section has been updated for clarity.
Are the results clearly presented? ( ) ( ) (x) ( ) The Results section has been carefully corrected and updated according to the reviewers’ comments.
Are the conclusions supported by the results? ( ) (x) ( ) ( ) The conclusions section has been revised to better reflect the results.
Are all figures and tables clear and well-presented? ( ) ( ) (x) ( ) Figures have been updated and are now clear and legible in the revised version.

The work presented here is an important practical aspect of fish farming, ensuring protection against microbial infections. The authors describe the beneficial effects of using extracts from the Solanaceae family as protective agents for fish. Given the growing antibiotic resistance of bacteria, research on the properties of other antibacterial agents is important in fish farming and human health.
However, I have a few comments for the authors regarding the work presented.

We thank the reviewer for the valuable feedback. Our detailed responses and corresponding edits are provided below and incorporated into the revised manuscript.

1) Regarding point 2.4. How long was the monitoring period for antibody levels after exposing the fish to the extracts?

The monitoring period for antibody response is till 28th day post immunization.
2) Regarding point 3.1.1: What differences were found in globulin levels after two weeks of HSF and WSF administration? Were there any significant differences between these groups? At what doses were these differences observed?

In the 2 weeks HSF administered groups, all the doses significantly elevated the globulin levels than the control (0 mg/kg). There was no difference between HSF and WSF groups.
3) Due to the small group of fish tested, how many repetitions of measurements were performed in each analysis? Were the results averaged from only six readings?

The sample sizes varied across assays, 6 fish for the immune parameters, and 10 fish for disease resistance assessments. Also, duplicate measurements were performed for each serum sample, while triplicate readings were conducted for the PBL (peripheral blood leukocyte) assays and ELISA analyses.
4) Regarding point 3.1.2, did WSF extracts increase lysozyme activity compared to HSF? At what concentration and time period were the greatest differences found? 

Overall, lysozyme activity did not differ significantly between WSF and HSF across most doses and time points. However, 1% WSF feeding for 1 week resulted in significantly higher lysozyme activity compared to HSF. This effect was not consistent at other concentrations or time periods.

5) Point 3.1.3. Please provide information on whether there were significant differences in anti-protease activity between WSF and HSF. Which compound improved this activity more effectively?

Anti-protease activity did not differ significantly between WSF and HSF. While both extracts exhibited moderate enhancement compared to the control, the effect could not be attributed more strongly to one extract over the other.

6) Regarding point 3.1.4: What is the reason for the large increase in values in graphs 4a and 4b after two and three weeks compared to one week? Was this sharp increase associated with a significant increase in ROS at specific doses between weeks? Were there any significant differences in the ROS levels generated by WSF and HSF in individual weeks and doses when compared directly?

The sharp increase in the values observed after two and three weeks reflects the kinetics of ROS generation: innate cellular responses peak after repeated stimulation and tend to show higher activity with prolonged exposure. This increase was not linked to statistically significant differences in ROS levels between WSF and HSF at individual doses and weeks. Rather, the effect appears to be dose- and time-dependent within each extract, without a consistent extract-level difference.

7) Regarding point 3.1.5, what caused the decrease in MPO activity after two and three weeks for both solutions?

The activation of neutrophils by the immunostimulant might triggered the degranulation of the MPO enzyme leads to decrease in intracellular MPO activity or ROS inactivated/ inhibited or damaged the MPO enzyme or the immunostimulants might enhanced only the non-neutrophil population, The induction of proliferation of leucocytes might be another reason, where the immature cells are reported to exhibit active ROS activity with low MPO activity. This may be an adaptive and self-regulative process of downregulating MPO to prevent excessive tissue damage from high ROS production upon the activation of NADPH oxidase. 
8) In the captions accompanying the graphs, the authors state that "errors with different letters indicate significant differences." It should be clarified whether this refers to bars within a week or between weeks, as this is sometimes unclear.

Thank you for your valuable suggestion. The suggested change has been done in the revised manuscript (line 225).
9) In Figures 7, 8, and 9, the letter markings indicating significance are illegible because the lines overlap too much and obscure the markings.

The figure is updated for a clear and legible representation.
10) Why was the observation period for the immune response extended to four weeks when the other analyses were conducted over three weeks?

This difference arises from the kinetics of the immune responses. In fish, which are cold-blooded vertebrates, the innate immune responses (both serum and cellular) typically peak within two weeks and can be readily measured within that time frame. In contrast, mounting a measurable antibody (adaptive) response generally requires a minimum of two weeks, and extending the observation to four weeks allows us to capture this response more reliably.

11) Lines 384-386: To what extent do differences in the level of protection depend on the method of administering plant extracts? Also, consider this study as an example.

The high relative percent survival rate was observed earlier in IP studies when compared to this oral administration. This discrepancy is likely due to several factors, such as, fish may not have ingested the optimal dosage, social hierarchy may result in competition for food and unequal access, digestion of immunostimulatory compounds resulted in degradation and poor absorption (line number: 400-402). Despite these limitations, oral administration of immunostimulants remains the preferred method for field applications, owing to its ease of use, non-stressful, less labour intensive and feasibility of large-scale administration. 
12) Did the authors check how long the protective properties beneficial to the body persist after discontinuing feed with WSF and HSF additives? For example, resistance to disease? Or, must continuous supplementation be maintained for these properties to occur?

We thank the reviewer for this valuable comment. In our study, fish were fed with each dose of WSF and HSF for up to three weeks. Following the feeding period, fish were subjected to pathogen challenge and monitored for 15 days. Across all experiments, the mid dose (0.1% feed) administered for two weeks produced the most consistent protective effect. These findings align with our previous studies, which also suggest that prolonged continuous supplementation does not necessarily improve outcomes and may place additional metabolic burden on the fish as well as unnecessary cost on the farmer. We have added this statement to the revised manuscript (lines 430-432)
13) Did the authors perform microbiological analyses of the WSF and HSF extracts at the analyzed concentrations to confirm their bactericidal effect?

Yes, we did study the bactericidal effect of these extracts on four major fish pathogens through inhibition zone assay and MTT assay to find out minimum inhibitory concentrations. But we did not include the results in this paper.

14) In general, the study lacks direct comparisons of the studied extracts. In the graphs, the authors refer to each extract separately, both in relation to the control and within concentrations. This is why I have questions about the direct differences between WSF and HSF; this is the only way to determine which is better, and the authors draw such conclusions.

We thank the reviewer for raising this point. Post hoc comparison showed no significant differences between WSF and HSF, though RPS values favoured 0.1% HSF at 2 weeks. As RPS is a derived measure, statistical testing was not applicable. WSF consistently fell short of HSF, likely due to the limited bioavailability of its polar compounds.

Reviewer 5 Report

Comments and Suggestions for Authors

The presented work is an important scientific study in the field of protection of Oreochromis mossambicus tilapia using a feed additive based on water-soluble (WSF) and hexane-soluble fractions (HSF) of extracts isolated from the leaves of Solanum trilobatum. The authors of the manuscript studied the qualitative composition of the compounds in the fractions of S. trilobatum extract and showed their effect on specific immunity, disease resistance, and humoral and cellular non-specific reactions in O. mossambicus. This research work is undoubtedly of scientific interest due to its novelty and relevance.

There are a number of comments that need to be changed:

1. In the “Simple Summary” section, the authors should provide the full names of the abbreviated terms, such as HSF, ROS, MPO, and WSF, for easier reading.

2. If possible, specify the qualitative composition of the 40% aromatic compounds and 11% steroids in the hexane-soluble fraction (HSF) of S. trilobatum. These aromatic and steroid compounds should be included in the table below the “Abstract” section.

3. Similarly, it is necessary to specify the qualitative composition of low-molecular-weight alcohols and carbonyls of the water-soluble fraction (WSF) of S. trilobatum. It is also advisable to provide these compounds in the table below the “Abstract” section.

4. In the “Introduction” section, when describing the plant S. trilobatum, it is advisable to indicate the general qualitative composition of alkaloids, flavonoids, tannins, and glycosides in parentheses in the text.

5. It is necessary to list in brackets the general qualitative composition of simple phenols, phenolic acids, isoflavones, xanthones, and lignans in S. trilobatum.

6. In the “Results” section, in subsections 3.1.4., 3.1.5., and 3.1.6., it is necessary to use the full names of the abbreviations - MPO and PBL.

7. There is no space before the reference [7] on line 73.

8. There are no periods at the end of the sentences on lines 68, 80, 127, and 144.

9. On line 97, the short name A. hydrophila should be used, as the authors have already provided the full name earlier in the text.

10. On line 109, the short name O. mossambicus should be used, as the authors have already provided the full name earlier in the text.

11. On line 117, the short name S. trilobatum should be used, as the authors have already provided the full name earlier in the text.

Author Response

Response to Reviewer 5 Comments

1. Summary

The presented work is an important scientific study in the field of protection of Oreochromis mossambicus tilapia using a feed additive based on water-soluble (WSF) and hexane-soluble fractions (HSF) of extracts isolated from the leaves of Solanum trilobatum. The authors of the manuscript studied the qualitative composition of the compounds in the fractions of S. trilobatum extract and showed their effect on specific immunity, disease resistance, and humoral and cellular non-specific reactions in O. mossambicus. This research work is undoubtedly of scientific interest due to its novelty and relevance.

There are a number of comments that need to be changed:

Thank you very much for taking the time to review this manuscript. Please find the detailed responses below and the corresponding revisions/corrections in track changes in the re-submitted files.

2. Questions for General Evaluation

Reviewer’s Evaluation

Does the introduction provide sufficient background and include all relevant references?

Yes

Are all the cited references relevant to the research?

Yes

Is the research design appropriate?

Yes

Are the methods adequately described?

Yes

Are the results clearly presented?

Yes

Are the conclusions supported by the results?

Yes

3. Point-by-point response to Comments and Suggestions for Authors

Comments 1: In the “Simple Summary” section, the authors should provide the full names of the abbreviated terms, such as HSF, ROS, MPO, and WSF, for easier reading.

Response 1: The necessary changes have been made in this section for better clarity.

Comments 2: If possible, specify the qualitative composition of the 40% aromatic compounds and 11% steroids in the hexane-soluble fraction (HSF) of S. trilobatum. These aromatic and steroid compounds should be included in the table below the “Abstract” section.

Response 2: Thank you for your valuable suggestion. Due to word limit constraints, we were unable to include this information in the abstract. However, all compounds have been comprehensively listed and categorized by chemical class in Tables 3.

Comments 3: Similarly, it is necessary to specify the qualitative composition of low-molecular-weight alcohols and carbonyls of the water-soluble fraction (WSF) of S. trilobatum. It is also advisable to provide these compounds in the table below the “Abstract” section.

Response 3: Thank you for your valuable suggestion. Due to word limit constraints, we were unable to include this information in the abstract. However, all compounds have been comprehensively listed and categorized by chemical class in Tables 2. As the journal’s formatting guidelines do not permit tables below the abstract, we omitted the table from that section.

Comments 4: In the “Introduction” section, when describing the plant S. trilobatum, it is advisable to indicate the general qualitative composition of alkaloids, flavonoids, tannins, and glycosides in parentheses in the text.

Response 4: Thank you for your valuable suggestion. We have added the information to the manuscript (lines 71-73).

Comments 5: It is necessary to list in brackets the general qualitative composition of simple phenols, phenolic acids, isoflavones, xanthones, and lignans in S. trilobatum.

Response 5: Thank you for your valuable suggestion. We have added the information to the manuscript (lines 71-73).

Comments 6: In the “Results” section, in subsections 3.1.4., 3.1.5., and 3.1.6., it is necessary to use the full names of the abbreviations - MPO and PBL.

Response 6: Thank you for your valuable suggestion. The suggested changes have been done in the revised manuscript.

Comments 7: There is no space before the reference [7] on line 73.

Response 7: Thank you for your valuable suggestion. The suggested change has been done in the revised manuscript.

Comments 8: There are no periods at the end of the sentences on lines 68, 80, 127, and 144.r

Response 8: Thank you for your valuable suggestion. The suggested changes have been done in the revised manuscript.

Comments 9: On line 97, the short name A. hydrophila should be used, as the authors have already provided the full name earlier in the text.

Response 9: Thank you for your valuable suggestion. The suggested change has been done in the revised manuscript.

Comments 10: On line 109, the short name O. mossambicus should be used, as the authors have already provided the full name earlier in the text.

Response 10: Thank you for your valuable suggestion. The suggested change has been done in the revised manuscript.

Comments 11: On line 117, the short name S. trilobatum should be used, as the authors have already provided the full name earlier in the text.

Response 11: Thank you for your valuable suggestion. The suggested change has been done in the revised manuscript.

Round 2

Reviewer 3 Report

Comments and Suggestions for Authors

The author has made point-to-point revisions according to the reviewer's suggestions. I agree to publish the revised manuscript in accordance with the journal format requirements.